# Gallocin A, an Atypical Two-Peptide Bacteriocin with Intramolecular Disulfide Bonds Required for Activity

Alexis Proutière,[a]* Laurence du Merle,[a] Marta Garcia-Lopez,[a] Corentin Léger,[b] Alexis Voegele,[b] Alexandre Chenal,[b] Antony Harrington,[c] ◉Yftah Tal-Gan,[c] Thomas Cokelaer,[d,e] Patrick Trieu-Cuot,[a] ◉Shaynoor Dramsi[a]

[a]Institut Pasteur, Université Paris Cité, CNRS UMR6047, Biology of Gram-Positive Pathogens Unit, Paris, France
[b]Institut Pasteur, Université Paris Cité, CNRS UMR3528, Biochemistry of Macromolecular Interactions Unit, Paris, France
[c]Department of Chemistry, University of Nevada, Reno, Reno Nevada, USA
[d]Institut Pasteur, Université Paris Cité, Plateforme Technologique Biomics, Paris, France
[e]Institut Pasteur, Université Paris Cité, Bioinformatics and Biostatistics Hub, Paris, France

**ABSTRACT** *Streptococcus gallolyticus* subsp. *gallolyticus* (*SGG*) is an opportunistic gut pathogen associated with colorectal cancer. We previously showed that colonization of the murine colon by *SGG* in tumoral conditions was strongly enhanced by the production of gallocin A, a two-peptide bacteriocin. Here, we aimed to characterize the mechanisms of its action and resistance. Using a genetic approach, we demonstrated that gallocin A is composed of two peptides, GllA1 and GllA2, which are inactive alone and act together to kill "target" bacteria. We showed that gallocin A can kill phylogenetically close relatives of the pathogen. Importantly, we demonstrated that gallocin A peptides can insert themselves into membranes and permeabilize lipid bilayer vesicles. Next, we showed that the third gene of the gallocin A operon, *gip*, is necessary and sufficient to confer immunity to gallocin A. Structural modeling of GllA1 and GllA2 mature peptides suggested that both peptides form alpha-helical hairpins stabilized by intramolecular disulfide bridges. The presence of a disulfide bond in GllA1 and GllA2 was confirmed experimentally. Addition of disulfide-reducing agents abrogated gallocin A activity. Likewise, deletion of a gene encoding a surface protein with a thioredoxin-like domain impaired the ability of gallocin A to kill *Enterococcus faecalis*. Structural modeling of GIP revealed a hairpin-like structure strongly resembling those of the GllA1 and GllA2 mature peptides, suggesting a mechanism of immunity by competition with GllA1/2. Finally, identification of other class IIb bacteriocins exhibiting a similar alpha-helical hairpin fold stabilized with an intramolecular disulfide bridge suggests the existence of a new subclass of class IIb bacteriocins.

**IMPORTANCE** *Streptococcus gallolyticus* subsp. *gallolyticus* (*SGG*), previously named *Streptococcus bovis* biotype I, is an opportunistic pathogen responsible for invasive infections (septicemia, endocarditis) in elderly people and is often associated with colon tumors. *SGG* is one of the first bacteria to be associated with the occurrence of colorectal cancer in humans. Previously, we showed that tumor-associated conditions in the colon provide *SGG* with an ideal environment to proliferate at the expense of phylogenetically and metabolically closely related commensal bacteria such as enterococci (1). *SGG* takes advantage of CRC-associated conditions to outcompete and substitute commensal members of the gut microbiota using a specific bacteriocin named gallocin, recently renamed gallocin A following the discovery of gallocin D in a peculiar *SGG* isolate. Here, we showed that gallocin A is a two-peptide bacteriocin and that both GllA1 and GllA2 peptides are required for antimicrobial activity. Gallocin A was shown to permeabilize bacterial membranes and kill phylogenetically closely related bacteria such as most streptococci, lactococci, and enterococci, probably through membrane pore formation. GllA1 and GllA2 secreted peptides are unusually long (42 and 60 amino acids long) and have very few charged amino acids compared

Address correspondence to Shaynoor Dramsi, shaynoor.dramsi@pasteur.fr, or Alexis Proutière, alexis.proutiere@epfl.ch.

*Present address: Alexis Proutière, Laboratory of Molecular Microbiology, Global Health Institute, School of Life Sciences, Ecole Polytechnique Fédérale de Lausanne (EPFL), Lausanne, Switzerland.

The authors declare no conflict of interest.

to well-known class IIb bacteriocins. *In silico* modeling revealed that both GllA1 and GllA2 exhibit a similar hairpin-like conformation stabilized by an intramolecular disulfide bond. We also showed that the GIP immunity peptide forms a hairpin-like structure similar to GllA1/GllA2. Thus, we hypothesize that GIP blocks the formation of the GllA1/GllA2 complex by interacting with GllA1 or GllA2. Gallocin A may constitute the first class IIb bacteriocin which displays disulfide bridges important for its structure and activity and might be the founding member of a subtype of class IIb bacteriocins.

**KEYWORDS** class IIb bacteriocin, antimicrobial peptides, immunity peptide, disulfide bond, bacteriocins, *Streptococcus gallolyticus*

*Streptococcus gallolyticus* subsp. *gallolyticus* (*SGG*), formerly known as *Streptococcus bovis* biotype I, is a gut commensal of the rumen of herbivores which causes infective endocarditis in elderly people and is strongly associated with colorectal cancer (CRC). In a previous study, we showed that *SGG* can take advantage of tumoral conditions (increased secondary bile salts concentration) to thrive and colonize the intestinal tract of Notch/APC mice. This colonization advantage was shown to be linked to the production of a two-component bacteriocin named gallocin, which enabled *SGG* to outcompete murine gut resident enterococci in tumor-bearing mice, but not in non-tumor mice (1). As such, gallocin constitutes the first bacterial factor explaining the association of *SGG* with CRC. The identification of a different gallocin, gallocin D, from the environmental isolate *SGG* LL009 (2) led to the renaming of *SGG* UCN34 gallocin as gallocin A.

Bacteriocins are highly diverse antimicrobial peptides secreted by nearly all bacteria. In Gram-positive bacteria, they are divided into three classes based on size, amino acid composition, and structure (3). Class I includes small (<10 kDa), heat-stable peptides that undergo enzymatic modification during biosynthesis; class II includes small (<10 kDa), heat-stable peptides without post-translational modifications; and class III includes larger (>10 kDa), thermolabile peptides and proteins. Class II bacteriocins are further subdivided into four subtypes: class IIa consists of pediocin-like bacteriocins, class IIb consists of bacteriocins with two peptides, class IIc consists of leaderless bacteriocins, and class IId encompasses all other non-pediocin-like, single-peptide bacteriocins with a leader sequence. Previous *in silico* analysis revealed that gallocin A, encoded by *gallo_2021* (renamed *gllA2*) and *gallo_2020* (renamed *gllA1*), belongs to the class IIb bacteriocins (Pfam10439) exhibiting a characteristic double glycine leader peptide. The third gene of this operon (*gallo_2019*, renamed *gip*) was thought to encode the immunity protein.

We previously showed that a secreted peptide, gallocin-stimulating peptide (GSP), activates transcription of the gallocin A core operon through a two-component system named BlpHR (4). The entire BlpHR regulon has been characterized and consists of 24 genes, 20 of which belong to the gallocin locus (4). Concomitantly, we showed that GSP, but also GllA1 and GllA2, are secreted by a unique ABC transporter named BlpAB (5). GllA1 and GllA2 are synthesized as pre-peptides with an N-terminal leader sequence cleaved during export after a double glycine motif to produce the extracellular mature active peptide. Well-known class IIb bacteriocins usually consist of two genes encoding short peptides, alpha and beta, which fold into alpha-helical structures and insert themselves into target bacterial membranes to alter their permeability, resulting in ion leakage and cell death (6).

The aim of this work was to characterize the gallocin A spectrum of activity, mode of action, and immunity mechanism. Our results indicate that the GllA1 and GllA2 peptides are atypical and contain a disulfide bond required for antibacterial activity. We showed that GllA1/GllA2 can permeabilize lipid bilayers. The predicted structure of the GIP immunity peptide strikingly mimics those of the GllA1 and GllA2 mature peptides, suggesting a mechanism of immunity by interference. *In vitro*, gallocin A was able to kill most closely related species, such as streptococci and enterococci, highlighting the potential of these narrow-spectrum antimicrobials as alternatives to antibiotics.

## RESULTS

**Gallocin A is a two-peptide bacteriocin.** As shown in Fig. 1A, the gallocin A core operon is composed of three genes (*gllA2*, *gllA1*, *gip*) which code for 2 putative bacteriocin peptides (GllA1 and GllA2) and a putative immunity protein (GIP). To demonstrate the role of *gllA1* and *gllA2* in gallocin A activity, we performed in-frame deletions of *gllA1* and *gllA2* separately in *SGG* strain UCN34 (wild-type, WT) and tested the antibacterial activity of the corresponding mutant supernatants by plate diffusion assays, as described previously (4). As shown in Fig. 1B, the antimicrobial activity of gallocin A is completely abolished in the supernatants of Δ*gllA1* and Δ*gllA2* mutants and is restored when the supernatants of Δ*gllA1* and Δ*gllA2* are combined in a 1:1 ratio. This demonstrates that both GllA1 and GllA2 are required for gallocin A activity and confirms that gallocin A is a two-peptide class IIb bacteriocin (3). Finally, we showed that gallocin A is active at a wide range of pH (2 to 12, Fig. S1A) and temperatures (Fig. S1B).

Because the gene encoding the putative immunity protein GIP cannot be deleted alone without self-intoxication of the bacterium, we used the original mutant UCN34Δ*blp* (1), in which the three genes of the gallocin A operon (*gllA2*, *gllA1*, and *gip*) were deleted, and tested its sensitivity to gallocin A. As expected, the Δ*blp* mutant became sensitive to gallocin A (Fig. 1C). Next, we complemented the Δ*blp* mutant with a plasmid encoding *gip* and showed that this was sufficient to restore bacterial growth of the recombinant strain in the presence of gallocin A. These results demonstrate that GIP confers immunity to gallocin A (Fig. 1C). Moreover, constitutive expression of *gip* in heterologous bacteria sensitive to gallocin (such as *Streptococcus agalactiae* and *Lactococcus lactis*) allowed their growth in the presence of gallocin (Fig. 1D). These results clearly demonstrate that expression of *gip* alone is necessary and sufficient to confer full immunity against gallocin A.

**Gallocin A is active against various streptococci and enterococci.** To further characterize the gallocin A activity spectrum, we tested the sensitivity of various bacteria from our laboratory collection, including species found as commensals in the gut and known Gram-positive human pathogens. We showed that gallocin A is active only against closely related bacteria, including various streptococci, enterococci, and lactococci, and was inactive against all other Gram-positive and Gram-negative bacteria tested (Fig. 2, Fig. S2A). Interestingly, the three different *S. agalactiae* strains tested (NEM316, BM110, and A909) differed significantly in their susceptibility to gallocin A. Similarly, the gallocin A sensitivity of many *Enterococcus faecalis* clinical isolates, including a few vancomycin-resistant isolates, was also variable (Fig. S2B). These results indicate that the gallocin A sensitivity of a given species can vary between strains.

**Gallocin A induces target cell-membrane depolarization.** To test whether gallocin A peptides can alter cell membrane permeability, as shown for well-studied class IIb bacteriocins, we assessed its impact on target cell membrane potential using the fluorescent voltage-dependent dye DiBAC4(3) [bis-(1,3-dibutylbarbituric acid)trimethine oxonol] and propidium iodide (PI). DiBAC4(3) can only access the cytoplasm when the membrane is depolarized, thus indicating an ion imbalance, and the DNA intercalator PI can only enter bacterial cells when the cytoplasmic membrane is compromised. The entry of PI and DiBAC4(3) into cells exposed to supernatants from UCN34 WT, Δ*blp* (no gallocin A), and Δ*blpS* (a mutant previously shown to overproduce gallocin A [4]) was assessed by flow cytometry. As shown in Fig. 3A and B, fluorescent dye penetration in *E. faecalis* OG1RF was increased in the presence of gallocin A compared to the control supernatant without gallocin A, indicating that gallocin A peptides can form pores in bacterial membranes.

It has been previously shown that pore formation by the two-peptide bacteriocins lactococcin G and enterocin 1071 requires the presence of UppP, a membrane protein involved in peptidoglycan synthesis that may serve as a receptor for these bacteriocins (7). To investigate whether gallocin A is active in the absence of a proteinaceous receptor, we tested its capacity to permeabilize lipid bilayer vesicles. To do so, we used large unilamellar vesicles (LUV) in which a fluorescence marker, 8-aminonaphthalene-1,3,6-trisulfonic acid (ANTS), and its quencher, p-xylene-bis-pyridinium bromide (DPX), are encapsulated. If pores are formed in the membrane of the liposomes, ANTS and DPX are released into the medium and ANTS recovers its fluorescence. As shown in Fig. 3C, the addition of UCN34

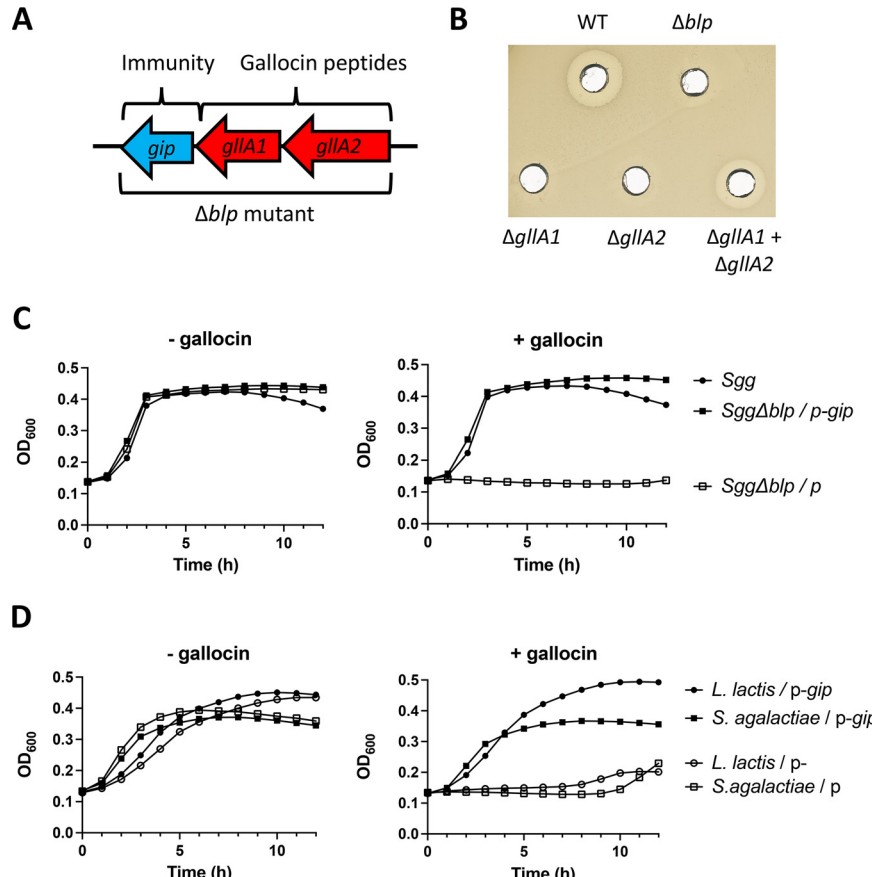

**FIG 1** Gallocin A is a two-peptide bacteriocin. (A) The core operon encoding gallocin A peptides and the immunity protein in *SGG* strain UCN34. Gallocin genes are indicated in red and renamed *gllA1* and *gllA2* according to the nomenclature of Hill et al. (2). (B) Agar diffusion assay to test gallocin activity from supernatants of UCN34 wild-type (WT), Δ*gllA1*, Δ*gllA2*, and Δ*blp* against gallocin-sensitive *Streptococcus gallolyticus* subsp. *macedonicus* (*SGM*) strain. One representative plate of three independent replicates is shown. (C and D) Growth curves of *SGG* Δ*blp*, *Streptococcus agalactiae* A909 and *Lactococcus lactis* NZ9000 containing an empty plasmid (p) or a plasmid expressing *gip* (p-*gip*) in Todd-Hewitt broth supplemented with 0.5% yeast extract (THY) medium supplemented with supernatant of Δ*blpS* (a strain overproducing gallocin A, "+ gallocin") or Δ*blp* (gallocin A deletion mutant, "– gallocin") and 0.01% of Tween 20. The mean of two independent replicates is shown.

WT supernatant containing gallocin A led to LUV permeabilization, while the supernatant of the Δ*blp* mutant had no effect, showing that gallocin A can alter the vesicle membrane. Of note, the addition of a small amount of Tween 20 (0.01%) was necessary to observe gallocin A activity. Importantly, the Δ*blp* supernatant supplemented with Tween 20 at 0.01% had no effect on liposomes, showing that the membrane permeabilization induced by the UCN34 WT supernatant is not caused by the detergent alone (Fig. 3C).

We also confirmed that both GllA1 and GllA2 were required for membrane permeabilization. Indeed, the addition of Δ*gllA1* or Δ*gllA2* supernatant alone had no effect, while the addition of both supernatants led to LUV permeabilization regardless of which peptide was added first (Fig. 3D).

**Gallocin A peptides contain a disulfide bond essential for their bactericidal activity.** Both GllA1 and GllA2 pre-peptides exhibit a typical N-terminal leader sequence of 23 amino acids, ending with 2 glycine residues, which is cleaved upon the secretion of these peptides through a dedicated ABC transporter (5). GllA1 and GllA2 mature peptides each contain 2 cysteines, which can potentially form a disulfide bridge important for their structure and function. Indeed, we showed that the addition of reducing agents such as dithiothreitol (DTT) or β-mercaptoethanol abolished gallocin A activity (Fig. 4A), whereas it has no effect on a control bacteriocin which does not possess a disulfide bond, such as nisin.

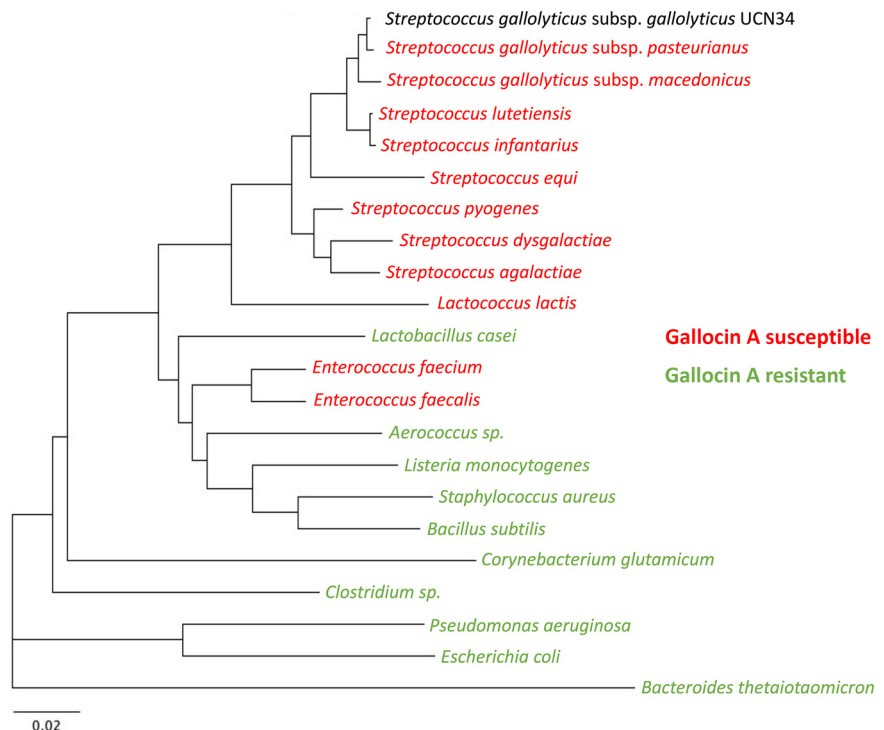

**FIG 2** Gallocin A is active against most streptococci, lactococci, and enterococci. Phylogenetic tree based on the 16S RNA sequence (from the Silva online database) of different bacterial species that are resistant (green) or susceptible (red) to gallocin A, as determined by agar diffusion assay (Fig. S2).

Furthermore, liquid chromatography-mass spectrometry (LC-MS) analysis provided the exact molecular masses of the mature GllA1 and GllA2 peptides. The calculated masses identified oxidized cysteine residues, indicating the presence of a disulfide bridge in each peptide (Fig. S3).

Interestingly, the gallocin A genomic locus in *SGG* UCN34 contains a conserved co-regulated gene (4), *gallo_rs10370*, which encodes a putative "bacteriocin biosynthesis protein" containing a thioredoxin domain (Fig. 4B). The thioredoxin domain is known to facilitate disulfide bond formation in *Escherichia coli* (8) and is predicted to be extracellular by Pfam/InterproScan. We hypothesized that this gene, renamed *blpT*, which encodes a surface protein potentially anchored to the cell wall, could assist disulfide bond formation in gallocin A peptides following secretion and cleavage of the leader peptide by the ABC transporter BlpAB (5). Indeed, deletion of this gene in UCN34 (Δ*blpT*) strongly altered the ability of *SGG* to outcompete *E. faecalis* OG1RF in competition experiments where attacker *SGG* and prey *E. faecalis* were inoculated together in Todd-Hewitt broth supplemented with 0.5% yeast extract (THY) liquid medium at a 1:1 ratio and counted on entero-agar plates after 4 h of coculture at 37°C (Fig. 4C). Remarkably, the Δ*blpT* mutant was comparable to the Δ*blp* mutant and WT revertant from *blpT* deletion (bWT) behaved like the parental UCN34 WT (Fig. 4C). Altogether, these results indicate that the existence of a disulfide bond in gallocin A mature peptides is important for their activity. Of note, the disulfide bond formation pathway of *E. coli*, containing the thioredoxin-like protein DsbA, was shown to be particularly important under anaerobic conditions (9). It is thus tempting to speculate that BlpT activity may be particularly important in the anaerobic environment that *SGG* encounters in the colon.

**The structural models of gallocin A peptides differ from those of other two-peptide bacteriocins.** Structural modeling of GllA1 and GllA2 pre- and mature forms was performed using ColabFold (10) and showed that the putative N-terminal leader sequences adopt disordered and extended conformations (Fig. 5A and B). The structural models of mature GllA1 and GllA2 are composed of two antiparallel alpha-helices,

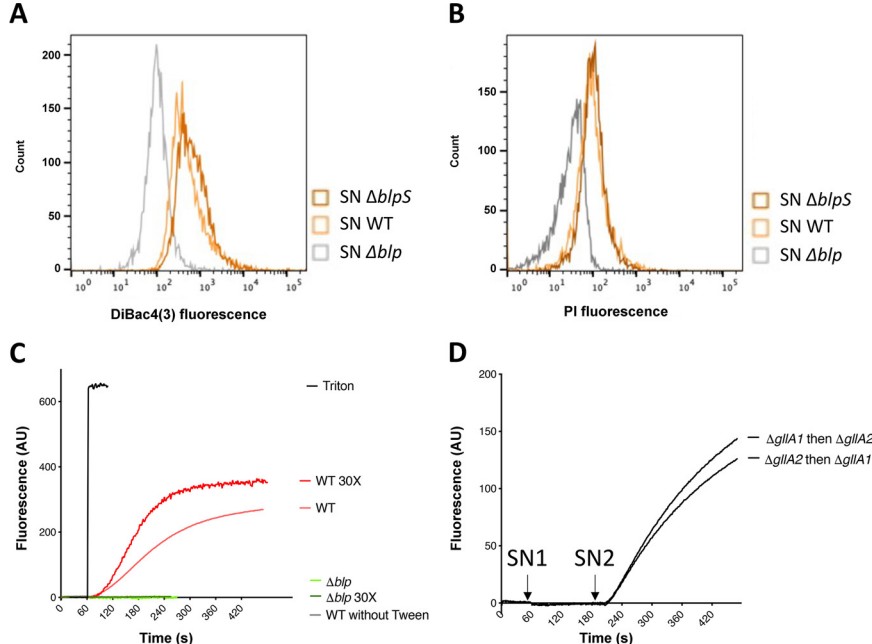

**FIG 3** Gallocin A can permeabilize bacterial membranes and lipid vesicles. (A and B) Fluorescence of the voltage-sensitive DiBAC4(3) [bis-(1,3-dibutylbarbituric acid)trimethine oxonol] (A) or membrane-impermeant propidium iodide (PI) (B) after resuspension of *Enterococcus faecalis* OG1RF in supernatants of UCN34 WT, Δ*blp* (– gallocin A), and Δ*blpS* (overexpressing gallocin A). One experiment representative of three independent replicates is shown. (C and D) Measure of the fluorescence corresponding to the release of ANTS (8-aminonaphthalene-1,3,6-trisulfonic acid; excitation: 390 nm, emission: 515 nm) encapsulated in large unilamellar vesicles after addition of *SGG* supernatant or Triton X-100 (positive control). (C) At 60 s, Triton or the supernatant of *SGG* UCN34 WT, Δ*blp*, WT 30× (concentrated 30 times), Δ*blp* 30×, was added to the liposomes. (D) At 60 s (SN1), the supernatant of Δ*gllA1* or Δ*gllA2* was added to the lipid vesicle suspension. At 200 s (SN2), the supernatant of the other strain was added. AU, arbitrary unit.

i.e., adopting an alpha-helical hairpin fold (Fig. 5A and B, Fig. S4A and B). Interestingly, the two cysteines of GllA1 and GllA2 face each other in each alpha-helix of the helical hairpins, forming an intramolecular disulfide bond. This suggests that the disulfide bonds in GllA1 and GllA2 reduce the conformational flexibility within each alpha-helical hairpin and stabilize their three-dimensional structures. Interestingly, modeling of the immunity peptide GIP shows striking structural similarities with the mature GllA1 and GllA2 peptides (Fig. 6A, Fig. S4C). Despite a relative low confidence (local distance difference test [lDDT] score between 50% and 65%), the five structural models of GllA1/GllA2, GllA1/GIP, and GllA2/GIP show similar orientations, giving credit to these models (Fig. 6B to D, Fig. S4D to F). As shown by aligning the Cα of each GIP in the GllA1/GIP and GllA2/GIP, we hypothesized that GIP intercalates between GllA1 and GllA2 (Fig. 6E). Thus, GIP may provide immunity by preventing the interaction between GllA1 and GllA2 within the cell membrane of the producing bacteria.

**Mechanisms of resistance to gallocin A.** To better understand the mode of action of gallocin A, we decided to investigate the mechanisms of resistance to gallocin A. For this purpose, we isolated 14 spontaneous resistant mutants (RSM-1 to -14) of the highly sensitive strain *S. gallolyticus* subsp. *macedonicus* (*SGM*) CIP105683T on agar plates supplemented with gallocin A (see Materials and Methods). As shown in Fig. S5B and C, 12 of these 14 mutants were able to grow in liquid THY medium supplemented with gallocin A, in contrast to the parental strain *SGM* WT. However, when grown in the presence of the control Δ*blp* supernatant, which does not contain gallocin A, all of the mutants exhibited a longer latency phase than the parental *SGM* WT, suggesting that the acquired mutations may have a fitness cost.

To identify the mutations conferring resistance to gallocin A in these mutants, we performed whole-genome sequencing using Illumina technology and compared their

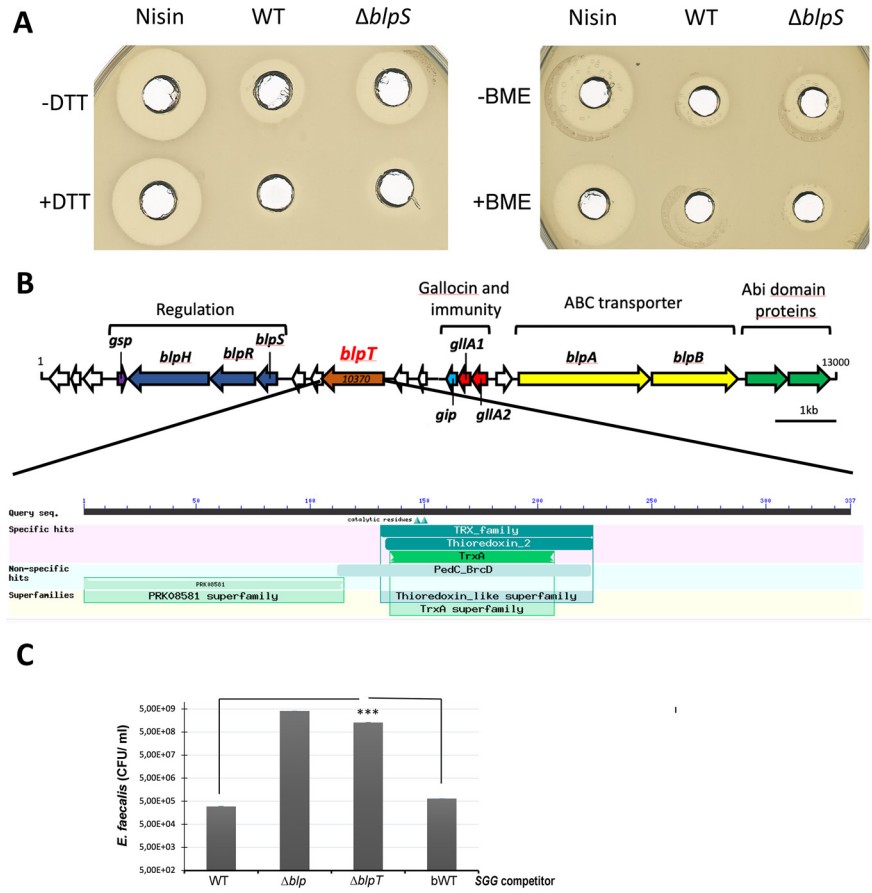

**FIG 4** Gallocin A peptides possess a disulfide bridge important for their structure and activity. (A) Agar diffusion assay to test bactericidal activity of purified nisin (25 μg/mL) and supernatants of *SGG* WT or Δ*blpS* supplemented or not with 50 mM dithiothreitol (DTT; left panel) or 100 mM β-mercaptoethanol (BME; right panel). One plate representative of three independent replicates is shown. (B) Schematic representation of the gallocin genomic locus and pBLAST domain identification in BlpT protein. (C) Recovered *E. faecalis* after coculture at a 1:1 ratio for 4 h with *SGG* WT, Δ*blp*, Δ*blpT*, and WT revertant from *blpT* deletion (bWT). The mean and standard deviation of three independent replicates is shown. Asterisks represent statistical differences with ***, $P < 0.001$ as assessed using two-way analysis of variance in GraphPad Prism version 9.

genomes with that of the parental strain that was *de novo* assembled using PacBio sequencing. Between 1 and 8 single nucleotide polymorphisms (SNPs)/deletions/insertions were identified in each RSM mutant compared to the WT controls (Table S1). Seven out of 12 mutants (RSM-1, RSM-2, RSM-4, RSM-5, RSM-6, RSM-12, RSM-14) had mutations in the genes encoding the WalKR two-component system (TCS) and 3 others (RSM-7, RSM-8, and RSM-10) had mutations in a gene (homologous to *gallo_rs1495*) encoding a putative "aggregation promoting factor" which contains a LysM peptidoglycan-binding domain and a lysozyme-like domain (Table S1, Fig. S6). The 2 remaining mutants (RSM-3 and RSM-11) displayed mutations which were not present in the other mutants and were located in other genes.

The WalRK TCS is known as the master regulator of cell wall homeostasis, cell membrane integrity, and cell division processes in Gram-positive bacteria (11). In streptococci, the response regulator WalR (VicR), but not the histidine kinase WalK (VicK), is essential. Consistent with this, the 2 mutations observed in WalR were single-amino acid substitutions (RSM-6, Ala$_{95}$ to Val; RSM-12, Arg$_{117}$ to Cys) while 4 out of the 5 mutations in WalK led to a frameshift or the appearance of a stop codon (Fig. S6).

Interestingly, three other mutants (RSM-7, RSM-8, and RSM-10) mapped in a single gene encoding a putative cell wall-binding protein with a C-terminal lysozyme-like domain. Two mutants (RSM-7 and RSM-8) exhibited frameshift mutations leading to the appearance

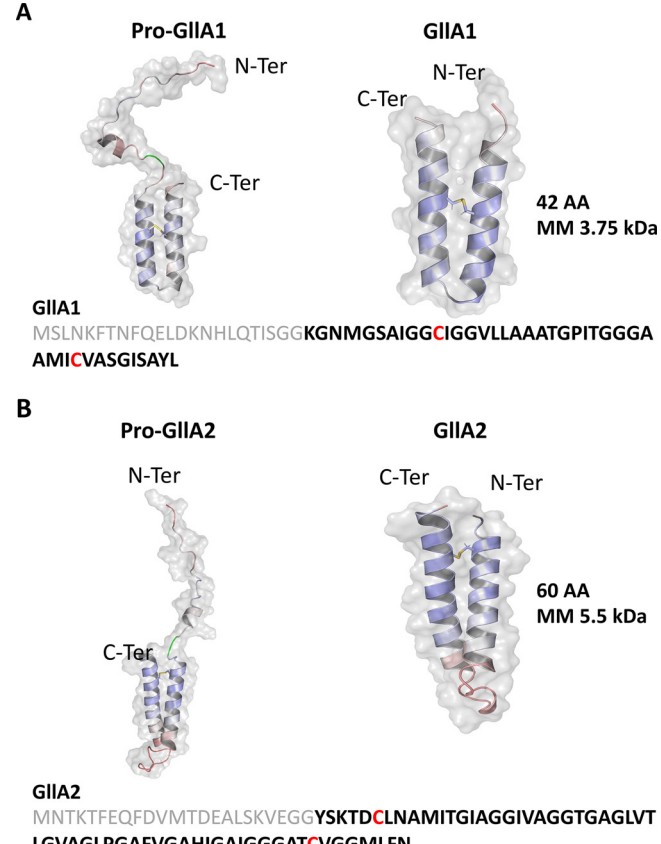

**FIG 5** Structural models of GllA1 and GllA2 predicted using ColabFold. (A and B) Pre-peptide and mature forms of GllA1 (A) and GllA2 (B) predicted using ColabFold, visualization was obtained with PyMOL (version 2.5.2 PyMOL Molecular Graphics System; Schrödinger, LLC). All representations are colored with a predicted local distance difference test (lDDT) score of 30% (red) to 100% (blue). For the pre-GllA1 and pre-GllA2, glycine doublet is colored in green. Disulfide bridges are represented as sticks.

of a premature stop codon, and the last one (RSM-10) exhibited a substitution of the putative key catalytic residue of the lysozyme-like domain ($E_{137}$ to K, Fig. S6).

Thus, we hypothesized that peptidoglycan alterations in these mutant strains could explain the resistance to gallocin A. To test this hypothesis, we labeled peptidoglycan with the fluorescent lectin wheat germ agglutinin (WGA-488) and imaged the mutants using conventional fluorescence microscopy. As shown in Fig. 7, most gallocin A-resistant mutants, including all WalKR mutants, exhibited abnormal morphology and formed small aggregates compared to the typical *SGM* WT linear chain of 2 to 5 cells. Cell morphology defects and peptidoglycan alterations were also detected in the 2 mutants which did not share common mutations with the other mutants (RSM-3 and -13, Fig. 7).

Taken together, these results suggest that alteration of the peptidoglycan structure can lead to gallocin A resistance, either by blocking its access to the membrane or by the formation of cell aggregates. It is worth noting that the RSM mutants' resistance to gallocin A was intermediate and that no potential membrane receptor for gallocin A peptides was identified.

## DISCUSSION

Gallocin A is a class IIb bacteriocin secreted by *S. gallolyticus* subsp. *gallolyticus* (*SGG*) to outcompete indigenous gut *E. faecalis* in tumoral conditions only (1). Mechanistically, gallocin A activity has been found to be enhanced by higher concentrations of secondary bile acids found under tumoral conditions (1). Another proof-of-concept study showed that *E. faecalis* carrying the conjugative plasmid pPD1 expressing bacteriocin was able to

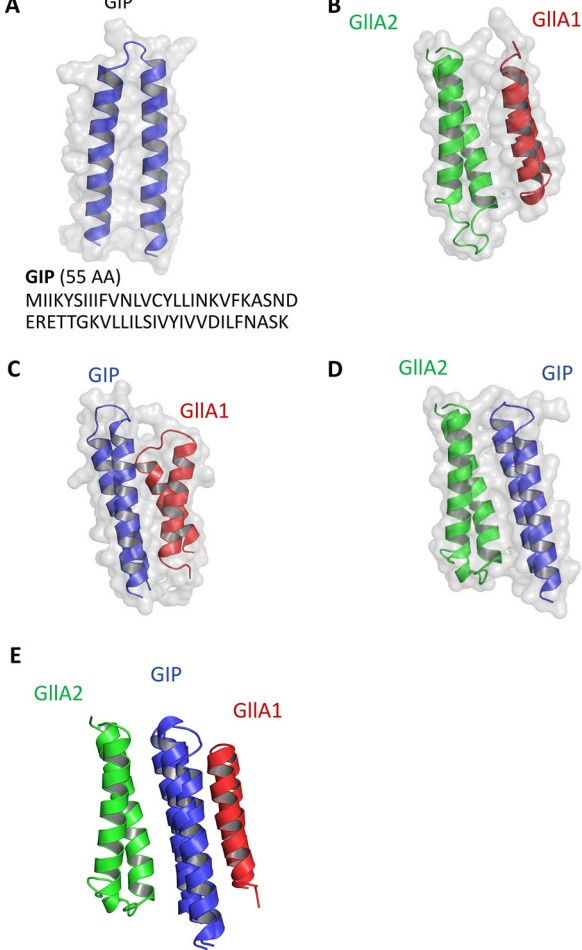

**A** GIP

**GIP (55 AA)**
MIIKYSIIIFVNLVCYLLINKVFKASND
ERETTGKVLLILSIVYIVVDILFNASK

**B** GIIA2  GIIA1

**C** GIP  GIIA1

**D** GIIA2  GIP

**E** GIIA2  GIP  GIIA1

**FIG 6** Structural models of GIP and its interactions with GIIA1 and GIIA2. (A) ColabFold modeling of GIP and visualization with PyMOL. (B to D) ColabFold modeling of the interaction between GIIA1/GIIA2 (B), GIP/GIIA1 (C), GIP/GIIA2 (D), and GIIA1/GIP/GIIA2. (E) Interaction models aligned on the $C\alpha$ of each GIP.

replace indigenous enterococci lacking pPD1 (11). The rise of antimicrobial resistance combined with the recognized roles of gut microbiota homeostasis in health has attracted renewed interest in the role of bacteriocins in gut colonization and their use as potential tools for editing and shaping the gut microbiome (12).

Here, we show that gallocin A, like many class IIb bacteriocins, only kills closely related species belonging to the *Streptococcaceae* and *Enterococcaceae* families. Interestingly, gallocin A can kill *Enterococcus faecium*, a commensal bacterium which greatly contributes to the transfer of antibiotic resistance in the microbiome and is classified as high priority in the WHO "priority pathogens list for R&D of new antibiotics." Taken together, these results highlight the potential of using bacteriocins such as gallocin A to fight antibiotic resistance and cure bacterial infections with a lower impact on the gut microbiota due to their narrow spectrum of action.

Both GIIA1 and GIIA2 are synthesized as pre-peptides with an N-terminal leader sequence which is cleaved during export after a GG motif via a specific ABC transporter, BlpAB, to produce the extracellular mature active peptides (5). Experimental determination of the molecular masses of GIIA1 and GIIA2 by LC-MS fits with a cleavage after the GG motif present in the leader sequence and indicates the presence of an intramolecular disulfide bond in GIIA1 and GIIA2. Moreover, reduction of these disulfide bonds abrogates gallocin A antimicrobial activity. ColabFold modeling of GIIA1 and GIIA2 indicates that the N-terminal leader sequence is unstructured and that the

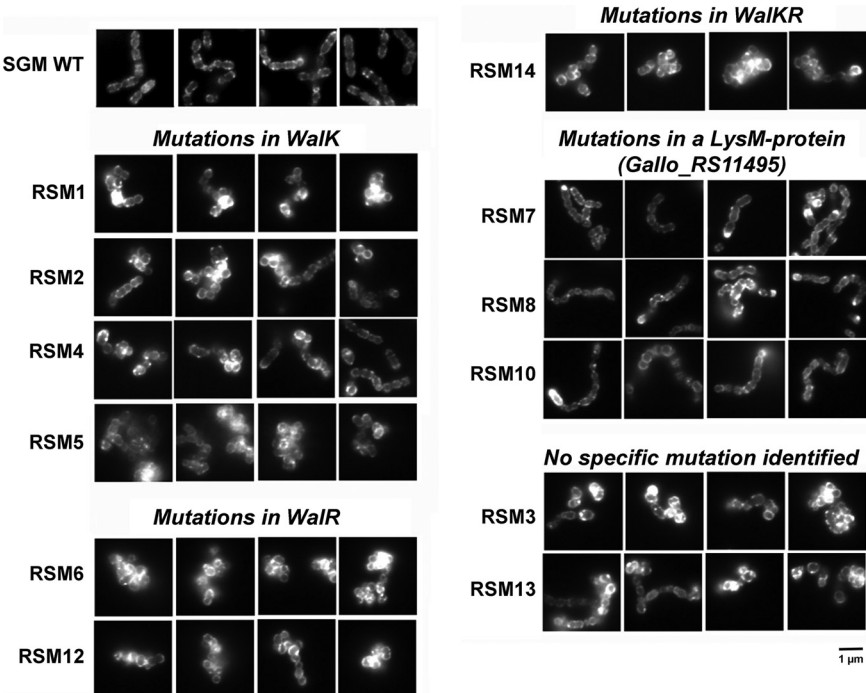

**FIG 7** Gallocin A-resistant mutants (RSM) form aggregates and exhibit morphological defects compared to the parental gallocin A-sensitive strain *SGM*. Epifluorescence microscopy images of *SGM* WT and RSM-1 to -14 (excluding the two mutants that did not grow in THY supplemented with gallocin A) labeled with the wheat germ agglutinin-488, a fluorescent peptidoglycan dye. Scale bar (1 $\mu$m) is shown on the bottom right. Representative images from three independent experiments are shown.

mature GIIA1 and GIIA2 share a similar structural fold with two antiparallel alpha-helices forming a hairpin stabilized by an intramolecular disulfide bond. To our knowledge, this is the first report of an intramolecular disulfide bond in class IIb bacteriocin peptides. Most class IIb peptides, including the well described lactococcin G, the plantaricin EF, the plantaricin JK, and the carnobacteriocin XY (CbnXY) (13–16), do not contain cysteine residues in their primary amino acid sequences. Consistently, the peptides constituting these 4 well-known bacteriocins are composed of only one main alpha-helix, and therefore do not require any disulfide bond to stabilize their three-dimensional structures. Recently, gallocin D was identified in a very peculiar strain, *SGG* LL009, isolated from raw goat milk in New Zealand (2). Gallocin D is a two-peptide bacteriocin homologous to infantaricin A secreted by *Streptococcus infantarius*, a member of the *Streptococcus bovis* group (2). Of note, the peptides of the 4 well-described two-peptide bacteriocins discussed previously and those of gallocin D are much smaller in size (about 30 amino acids long) than the gallocin A peptides (2). In addition, gallocin A peptides are less positively charged (1 positively charged amino acid in GIIA1, 2 in GIIA2), while the highly positively charged C terminus of lactococcin G $\alpha$-peptide is thought to contribute to the anchoring of the peptide to the membrane thanks to the transmembrane potential (negative inside) (13, 17).

A few other class IIb bacteriocins, such as brochocin C, thermophilin 13, and ABP-118 (18–21), were found to share similar structural properties with gallocin A peptides (longer peptides, few positively charged amino acids, and two cysteine residues in each peptide located close to the N/C terminus). AlphaFold modeling of these peptides showed that their putative structures resemble those of GIIA1 and GIIA2, with two antiparallel alpha-helices. Disulfide bonds between the cysteines of the 2 helices were also predicted in 5 out of the 6 peptides (Fig. S7). BrcB, the peptide without a predicted disulfide bond, was also the one with the worst lDDT score, suggesting that the prediction may not be accurate. In conclusion, gallocin A, as well as other class IIb

bacteriocins such as brochocin C, thermophilin 13, and ABP-118, may represent a subgroup which differs in structure, and potentially in their mode of action, from other well-known class IIb bacteriocins.

Finally, gallocin A resistance was studied through whole-genome sequencing of 12 spontaneous resistant mutants derived from the highly sensitive strain *S. gallolyticus* subsp. *macedonicus* CIP105683T. Previously, this method allowed the identification of UppP as a membrane receptor required for lactococcin G activity (7). Unlike this previous study, we did not find a common gene mutated in our 12 resistant mutants, suggesting that gallocin A does not require a specific receptor. This is in agreement with our data showing that gallocin A can permeabilize lipid vesicles composed of two phospholipids (phosphatidylcholine and phosphatidylglycerol). The majority of RSM mutants exhibited mutations in the genes encoding a regulatory two-component system sharing strong homologies with WalKR (also known as VicKR and YycGF). This two-component system, originally identified in *Bacillus subtilis*, is very highly conserved and specific to low GC% Gram-positive bacteria, including several pathogens such as *Staphylococcus aureus* (22, 23). Several studies have unveiled a conserved function for this system in different bacteria, including several streptococcal pathogens, defining this signal transduction pathway as a master regulatory system for cell wall metabolism (23). Consistent with the potential defect in cell wall synthesis, these mutants showed morphological abnormalities and cell division defects. Similar observations have been reported in *S. aureus* (24–26), where mutations in *walK* were shown to confer intermediate resistance to vancomycin and daptomycin.

Three mutants displayed independent mutations in a small protein (197 amino acids) of unknown function containing an N-terminal LysM peptidoglycan-binding domain and a C-terminal lysozyme-like domain. The lysozyme-like domain, which is about 50 amino acids long, was originally identified in enzymes that degrade bacterial cell walls. Interestingly, the mutations in the RSM-7, RSM-8, and RSM-10 mutants all mapped within the lysozyme-like domain, suggesting a potential alteration of the cell wall in these mutants. Finally, the last two last mutants (RSM-3 and RSM-13), carrying mutations in genes other than *walRK*, exhibited the same morphology defects associated with gallocin A resistance.

To conclude, it is worth noting that the 12 mutants were only partially resistant to gallocin A. Most RSM mutants formed bacterial aggregates which probably contributed to their gallocin A resistance, just as biofilms are more resistant to antibiotics. No specific membrane receptor could be identified for gallocin A. Interestingly, it has also been suggested that thermophilin 13, another class IIb bacteriocin that shares putative structural similarity with gallocin A (18), does not require any specific receptor for its activity. However, the different level of susceptibility to gallocin A within a given species, as demonstrated for three group B *Streptococcus* strains (A909 > BM110 > NEM316), as well as its narrow-spectrum mode of action, indicate that unidentified bacterial factors can modulate gallocin A sensitivity. In the future, it will also be important to identify the direct bacterial targets of gallocin A in the murine colon using global 16S DNA sequencing under normal and tumoral conditions.

## MATERIALS AND METHODS

**Cultures, bacterial strains, plasmids, and oligonucleotides.** The streptococci and enterococci used in this study were grown at 37°C in Todd-Hewitt broth supplemented with 0.5% yeast extract in standing filled flasks. When appropriate, 10 $\mu$g/mL of erythromycin was added for plasmid maintenance.

Plasmid construction was performed by PCR amplification of the fragment to be inserted with Q5 High-Fidelity DNA polymerase (New England Biolabs), digestion with the appropriate FastDigest restriction enzymes (Thermo Fisher Scientific), ligation with T4 DNA ligase (New England Biolabs), and transformation in commercially available TOP10-competent *E. coli* (Thermo Fisher Scientific). *E. coli* transformants were cultured in Miller's LB medium supplemented with 150 $\mu$g/mL erythromycin (for pG1-derived plasmids) or 50 $\mu$g/mL kanamycin (for pTCV-derived plasmid). Verified plasmids were electroporated in *S. agalactiae* NEM316 and mobilized from NEM316 to *SGG* UCN34 by conjugation as described previously (27). pTCV-derived plasmids were electroporated in *L. lactis* NZ9000. The strains, plasmids, and primers used in this study are listed in Tables S2 and S3. The wide range of bacteria

tested *in vitro* for their resistance or sensitivity to gallocin A antimicrobial activity came from our laboratory repository and were cultured in their optimal media and conditions.

**Construction of markerless deletion mutants in *SGG* UCN34.** In-frame deletion mutants were constructed as described previously (27). Briefly, the 5′ and 3′ regions flanking the region to be deleted were amplified and assembled by splicing by overlap extension PCR and cloned into the thermosensitive shuttle vector pG1. Once transformed in UCN34, the cells were cultured with erythromycin at 38°C to select for chromosomal integration of the plasmid by homologous recombination. About 4 single-crossover integrants were serially passaged at 30°C without antibiotic to facilitate the second event of homologous recombination and excision of the plasmid resulting either in gene deletion or bWT. In-frame deletions were identified by PCR and confirmed by DNA sequencing of the chromosomal DNA flanking the deletion.

**Gallocin A production assays.** Briefly, one colony of the indicator strain, *Streptococcus gallolyticus* subsp. *macedonicus*, was resuspended in 2 mL THY, grown until exponential phase, and poured onto a THY agar plate; the excess liquid was removed, and the plate was left to dry under the hood for about 20 min. Using sterile tips, 5-mm-diameter wells were dug into the agar. Each well was then filled with 80 $\mu$L of filtered supernatant from 5-h cultures (stationary phase) of *SGG* UCN34 WT or otherwise isogenic mutant strains and supplemented with Tween 20 at a final concentration of 0.1%. Inhibition rings around the wells were observed the following morning after overnight incubation at 37°C.

**Competition experiments.** *SGG* strains were inoculated from fresh agar plates at an initial OD$_{600}$ (optical density at 600 nm) of 0.1 together with *E. faecalis* OG1RF in THY medium and incubated for 4 h at 37°C in microaerobiosis. After 4 h of co-culture, the mixed cultures were serially diluted and plated onto *Enterococcus* agar-selective plates (BD Difco). On these plates, *SGG* exhibits a pale pink color while *E. faecalis* exhibits a strong purple color. CFU were counted the next morning to determine the final concentration in CFU/mL for each test sample.

**Analysis of gallocin A peptides by LC-MS.** *SGG* UCN34 was grown in 500 mL of sterile THY supplemented with 5 nM synthetic GSP at 37°C with 5% CO$_2$ for 12 to 16 h. The cultures were centrifuged at 4,000 × *g* for 20 min and the supernatant was filtered through a sterile 0.22-$\mu$m polyethersulfone (PES) filter. Ammonium sulfate was added to the filtered supernatants to give a concentration of 20% (wt/vol) and mixed by inversion until all ammonium sulfate salts went into the solution. The solution was stored at 4°C for 1 h, followed by centrifugation at 4,000 × *g* for 20 min. The supernatants were discarded, and the remaining pellet was dissolved in 100 mL deionized (DI) water and placed in a 3-kDa molecular weight cutoff (MWCO) dialysis tube. The dialysis tube was placed in a 500 mL graduated cylinder containing distilled water and a stir bar. Dialysis was performed for 4 h with DI water changed every hour. The material in the dialysis tube was then lyophilized. A 5-mg/mL solution of the lyophilized material was prepared in 75:25 (H$_2$O:ACN [acetonitrile]) and 50 $\mu$L was injected into an Agilent Technologies 6230 time-of-flight mass spectrometer (an HRMS system) with the following settings for positive electrospray ionization (ESI+) mode: capillary voltage = 3,500 V; fragmentor voltage = 175 V; skimmer voltage = 65 V; Oct 1 RF Vpp = 750 V; gas temperature = 325°C; drying gas flow rate = 0.7 L/min; nebulizer; 25 lb/in²; acquisition time = 17.5 min. An XBridge C$_{18}$ column (5 $\mu$m, 4.6 × 150 mm) was used for the LC-MS analysis.

**Membrane permeabilization assays.** These assays were performed as described previously (28). Briefly, ANTS (fluorophore probe) and DPX (quencher) were encapsulated into LUVs to monitor membrane permeabilization induced by peptides. The LUVs were prepared at a concentration of 10 mM lipid at a POPC:POPG (1-palmitoyl-2-oleoyl-sn-glycero-3-phosphocholine:1-palmitoyl-2-oleoyl-sn-glycero-3-phospho-(1′-rac-glycerol)) molar ratio of 8:2 containing 20 mM ANTS and 60 mM DPX. The multilamellar vesicle suspension was extruded through 0.4- and 0.2-$\mu$m polycarbonate filters to produce LUVs 200 ± 30 nm in diameter, as measured by dynamic light scattering. Unencapsulated ANTS and DPX were removed by gel filtration through a 5-mLSephadex G-25 column (Cytiva). For permeabilization assays, LUVs were incubated in buffer at 0.45 mM lipids at 25°C in a 101-QS cuvette (Hellma, France) and under constant stirring. The excitation wavelength was set to 390 nm and the emission of ANTS was continuously measured at 515 nm. The maximum intensity of permeabilization, corresponding to the maximum recovery of ANTS fluorescence, was measured after addition of 0.12% (2 mM) of Triton X-100.

**Generation of gallocin-resistant mutants.** To generate gallocin-resistant mutants, we concentrated *SGG* supernatant 200 times by precipitation with 20% ammonium sulfate. By serial 2-fold dilutions, we showed that this supernatant was approximately 64 times more concentrated than the original supernatant (Fig. S5A). Fourteen resistant mutants (named RSM-1 to -14) of the *S. gallolyticus* subsp. *macedonicus* parental strain CIP105683T, the species showing the highest sensitivity to gallocin A, were selected on THY agar plates containing 10% of concentrated supernatant. Twelve of these were confirmed to be gallocin-resistant by growth in THY supplemented with the supernatant of *SGG* WT containing gallocin and 0.01% Tween 20, which is necessary for gallocin A activity (Fig. S4B and C). As an important control, the same experiment was performed after precipitation of the Δ*blp* supernatant, which does not produce gallocin A. *SGM* WT was re-isolated on this plate and a single colony was stocked and sequenced with the RSM mutants as described below.

**Sequencing and SNP localization.** Whole-genome sequencing of the control *SGM* WT, re-isolated from Δ*blp* plate as described previously, and RSM mutants was performed using Illumina technology and compared with the genome of the parental strain *SGM* CIP105683T which was *de novo* assembled using PacBio sequencing. This assembly was performed with Canu v1.6 (29), leading to a main chromosome of 2,210,410 bp and a plasmid of 12,729 bp (PRJNA940176). The annotation was subsequently made with Prokka (30) before variant calling was performed using the Sequana (31) variant-calling pipeline. Of note, variants were called with a minimum frequency of 10% and a minimum strand balance of 0.2. Many mutations, probably due to the different method used to sequence the reference sequence,

were present in the control *SGM* WT strain and the RSM mutants. Therefore, only RSM-specific mutations occurring at a frequency of >0.5 compared to the control *SGM* WT were considered for this analysis and are shown in Table S1.

## SUPPLEMENTAL MATERIAL

Supplemental material is available online only.

**SUPPLEMENTAL FILE 1**, PDF file, 13.5 MB.

**SUPPLEMENTAL FILE 2**, PDF file, 0.3 MB.

## ACKNOWLEDGMENTS

We especially thank Tarek Msadek for careful and critical reading of the manuscript.

This study has been funded by the Institut National Contre le Cancer (INCA) PLBIO 16-025, attributed to S.D., and by the French Government's Investissement d'Avenir program, Laboratoire d'Excellence "Integrative Biology of Emerging Infectious Diseases (grant no. ANR-10-LABX-62-IBEID)". This study was also funded by the National Science Foundation (NSF, no. CHE-1808370 to Y.T.-G.).

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
