## [Reviewer comments · Microbiology Spectrum]

Microbiology Spectrum

Gallocin A, an atypical two-peptide bacteriocin with intramolecular disulfide bonds required for activity

Alexis Proutière, Laurence du Merle, Marta Garcia-Lopez, Corentin Léger, Alexis Voegele, Alexandre Chenal, Anthony Harrington, Yftah Tal-Gan, Thomas COKELAER, Patrick Trieu-Cuot, and Shaynoor Dramsi

Corresponding Authors: Shaynoor Dramsi and Alexis Proutière, Institut Pasteur

Review Timeline:

Submission Date:	December 11, 2022
Editorial Decision:	January 13, 2023
Revision Received:	February 13, 2023
Accepted:	February 16, 2023

Editor: Christopher LaRock

Reviewer(s): The reviewers have opted to remain anonymous.

Transaction Report:

DOI: <https://doi.org/10.1128/spectrum.05085-22>

January 13, 2023

Dr. Shaynoor Dramsi
Institut Pasteur
28 rue du Dr Roux
Paris
France

Re: Spectrum05085-22 (Gallocin A, an atypical two-peptide bacteriocin with intramolecular disulfide bonds required for activity)

Dear Dr. Shaynoor Dramsi:

On the basis of recommendations from expert reviewers in the field, I have determined that your manuscript requires edits before acceptance. The reviewers found the study to overall be of interest to the field, but noted several points for clarification or where additional information is needed to fully support the claims and improve presentation. While you are revising the manuscript, please specifically address the comments copied below from the Reviewers. Keep in mind to all claims must be experimentally supported, so avoid speculation or interpretation beyond that which is specifically demonstrated in an experiment. Please also carefully check grammar and other writing throughout, referring to journal guidelines with regards to all formatting requirements.

Thank you for submitting your manuscript to Microbiology Spectrum. As you will see your paper is very close to acceptance. Please modify the manuscript along the lines I have recommended. As these revisions are quite minor, I expect that you should be able to turn in the revised paper in less than 30 days, if not sooner. If your manuscript was reviewed, you will find the reviewers' comments below.

When submitting the revised version of your paper, please provide (1) point-by-point responses to the issues raised by the reviewers as file type "Response to Reviewers," not in your cover letter, and (2) a PDF file that indicates the changes from the original submission (by highlighting or underlining the changes) as file type "Marked Up Manuscript - For Review Only". Please use this link to submit your revised manuscript. Detailed instructions on submitting your revised paper are below.

Link Not Available

Sincerely,

Christopher LaRock

Reviewer comments:

Reviewer #1 (Comments for the Author):

This paper is a very nice study of a novel type of class IIB bacteriocin possessing disulfide bonds, which the authors have named Gallocin A. Despite the lack of purified Gallocin A in their assays, they have provided appropriate controls for their experiments that I believe provide good evidence for their findings. I have a few comments for the author.

- 1) You mention that the strains of *S. agalactiae* you tested have differences in susceptibility to gallocin A. Based on the finding that the mechanism of action of gallocin is likely membrane permeabilization, do you think this is due to differences in membrane composition, or do you think that this is due to some other process? Is it possible that multiple mechanisms of gallocin killing could exist, similar to what has been found for some other bacteriocins?
- 2) Do the authors know or have a hypothesis as to why Tween 20 was necessary for gallocin A activity in their vesicle membrane model?
- 3) Please show the data for the β -mercaptoethanol abolishing gallocin A activity, or take the data not shown out of the paper.

- 4) This is probably a bit outside the scope of the paper, as DTT/BME is shown to abrogate gallocin A activity in plates, but does mutation of the Cys in galA1/galA2 result in inability of the peptides to have antibacterial activity? In the same line, does deletion of blpT result in the lack of the oxidized disulfide bond observed by LC-MS/MS in galA1/galA2?
- 5) From the evidence shown, it appears that blpT is likely important in disulfide bond formation in galA1/galA2. Following this, if disulfide bond formation is important for galA1/galA2 activity, does deletion of blpT result in a phenotype similar to deletion of blp in the plate assays?
- 6) I know that purification of these types of peptides can be difficult, but can you comment on if you have examined isolation of these, and examined if they interact directly by ITC or the like?
- 7) Do you have an idea of how much gallocin A is produced by the WT strain?
- 8) You mention that the WalRK mutants exhibit abnormal morphology, this is clearly shown in Fig. 7 and is well documented in the literature for other Streptococci. Could you provide enumeration of the chain length in SGM WT vs. these resistant mutants, i.e. are they statistically different? One of two of your more severe mutants would be acceptable.
- 9) You mention in your methods that your control SGM WT strain had mutations compared to the reference sequence, and the RSM mutants did as well. Are these background mutations shared between WT and RSM mutants? Do you believe this is due to strain mutation/adaptation over time in the laboratory, as has been observed for other species?

Other minor comments:

- Some of the pictures in Fig. S2 are quite pixelated. Could you provide higher resolution pictures for these? Also it would benefit to have multiple images, although I know this could make the figure quite busy and I don't think is absolutely necessary.
- In Figure S5, control is spelled in French. Please change to English.
- In Line 254, I believe the authors meant to refer to Table 2 instead of Table 1.
- Figure 4C, please outline what statistical test is used to compare the conditions.
- Figure S5 is a little difficult to interpret, Could you present in a slightly different way to make the areas of mutation larger?
- Please add scale bars to Figure 7 for the microscopy.
- Line 423-OD is spelled incorrectly.
- Can you outline how many times each of the experiments were performed in the figure legends?

Reviewer #2 (Public repository details (Required)):

Bacterial genome data

Reviewer #2 (Comments for the Author):

General comments:

The authors studied the function of gallocin A, a bacteriocin of *Streptococcus gallolyticus* subsp. *gallolyticus*. The aim of the study is to characterize the mode of action and the mechanisms of resistance for gallocin A. The authors can demonstrate lipid bilayer disruption by gallocin A and they show by genetic approaches that gallocin A is a two peptide bacteriocin and that GIP functions as an immunity protein. The main novelty of the manuscript are the disulfide bonds proposed to be important for the bacteriocin activity. This hypothesis is confirmed by a loss of bacteriocin activity upon treatment with reducing agents and the detection of a thioredoxin domain in a gene of the bacteriocin cluster that upon knockout causes diminished bacteriocin activity. To substantiate this observation a mutation of cysteine residues of gallocin A should be performed. Some information about the concentration at which gallocin A is killing bacterial target strains should also be included in the manuscript. Analysis of gallocin A resistant mutants suggest an involvement of the cell wall structure in the resistance mechanism but could not identify a specific protein receptor.

Specific comments:

1 Page 2 line 55

SGG is not only associated with asymptomatic colon tumors, please modify.

2 Page 12 line 297-304

Here the concentrations at which gallocin is active should be mentioned and that the spectrum of activity against closely related species is in line with previous investigations should be added.

3 Page 24 line 624

Legend of Fig 4A Please provide information on the concentration of nisin that was used in this assay.

4 Fig. 2:

While the sensitivity of *S. agalactiae* has been tested. What about other pathogenic streptococci, like *S. pyogenes* or *S. dysgalactiae* subsp *equisimilis*?

Preparing Revision Guidelines

Please return the manuscript within 60 days; if you cannot complete the modification within this time period, please contact me. If you do not wish to modify the manuscript and prefer to submit it to another journal, please notify me of your decision immediately so that the manuscript may be formally withdrawn from consideration by Microbiology Spectrum.

Reviewer comments:

Reviewer #1 (Comments for the Author):

This paper is a very nice study of a novel type of class IIB bacteriocin possessing disulfide bonds, which the authors have named Gallocin A. Despite the lack of purified Gallocin A in their assays, they have provided appropriate controls for their experiments that I believe provide good evidence for their findings. I have a few comments for the author.

We thank you for your positive comment on our manuscript.

1) You mention that the strains of *S. agalactiae* you tested have differences in susceptibility to gallocin A. Based on the finding that the mechanism of action of gallocin is likely membrane permeabilization, do you think this is due to differences in membrane composition, or do you think that this is due to some other process?

This is a very good but very difficult question to answer. In the liposome assay, we do observe a direct membrane permeabilization by gallocin A. But access to the GBS membrane in living bacteria can be limited or promoted by strain-specific peptidoglycan associated factors and/or capsule composition. Besides that, we cannot discard the possibility suggested by the Reviewer about differences in membrane composition.

Is it possible that multiple mechanisms of gallocin killing could exist, similar to what has been found for some other bacteriocins?

To our knowledge, all class IIb bacteriocins kill their target cells through membrane permeabilization. This is also what we have found for gallocin A. We cannot exclude the possibility of other mechanisms of action.

2) Do the authors know or have a hypothesis as to why Tween 20 was necessary for gallocin A activity in their vesicle membrane model?

Class IIb bacteriocin are known to adopt their tri-dimensional structure when exposed to hydrophobic environment (see Nissen-Meyer et al., 2010) and Tween 20 may provide this environment. We think that addition of small amount of Tween 20 in the supernatant helps to disperse GIIA1/GIIA2 complexes, thus allowing each peptide of Gallocin A to reach the target cell membrane and refold in their active state once in the membrane.

3) Please show the data for the β -mercaptoethanol abolishing gallocin A activity, or take the data not shown out of the paper.

We have added the data showing the effect of β -mercaptoethanol on gallocin A activity.

4) This is probably a bit outside the scope of the paper, as DTT/BME is shown to abrogate gallocin A activity in plates, but does mutation of the Cys in galA1/galA2 result in inability of the peptides to have antibacterial activity?

We agree with the reviewer that it would be a very good experiment to prove the importance of the cysteine residues in the gallocin A peptides. However, we are missing a tool to detect gallocin A peptides. All our attempts to tag the gallocin peptides abrogated the peptide activity. Thus, we have no simple way to check if the mutated GIIA1/GIIA2 peptide would be produced to the same quantity and not degraded in the presence of this mutation. Moreover, constructing mutants in *Streptococcus gallolyticus* is still quite challenging and time-consuming. We thought about introducing the mutations on the complementation plasmid, however since complementation is already only partial with the wild-type gene, it will be difficult to get unambiguous results.

In the same line, does deletion of blpT result in the lack of the oxidized disulfide bond observed by LC-MS/MS in galA1/galA2?

We have not sent Δ blpT supernatant for LC-MS/MS determination for the reasons stated in point 5 (see below).

5) From the evidence shown, it appears that blpT is likely important in disulfide bond formation in galA1/galA2. Following this, if disulfide bond formation is important for galA1/galA2 activity, does deletion of blpT result in a phenotype similar to deletion of blp in the plate assays?

This is a very good question. Deletion of *blpT* did not produce any phenotype on plate assays while it had a striking effect in direct competition assay. We hypothesize that addition of Tween 20 and presence of oxygen could help spontaneous disulfide bond formation (Meehan et al., 2017, J Bacteriol.) during plate assays. We thus decided to perform the plate assay in anaerobic conditions, and we did observe reduction of gallocin activity in the *blpT* mutant as compared to the wild-type, but replicates were not always consistent. So, we decided not to show these results. Similarly, addition of a small amount of Tween 20 (0.1%) in the *blpT* mutant during direct competition assays in liquid medium increased killing activity of gallocin A but these results not being fully conclusive, we left them out.

LC/MS-MS on the Δ *blpT* supernatant is a great idea but should be performed in full anaerobic conditions, which remain a technical challenge and beyond the scope of the paper.

6) I know that purification of these types of peptides can be difficult, but can you comment

on if you have examined isolation of these, and examined if they interact directly by ITC or the like?

Several experiments were carried out to purify these peptides but none of them was successful

- Addition of a hemagglutinin or 6XHis tag to the C/N-terminal end of gIIA1/gIIA2 peptides resulted in loss of gallocin A activity.
- Chemical synthesis of GIIA1 and GIIA2 peptides failed.
- Fusion of GIIA1 and GIIA2 to Glutathion S-transferase (GST) protein to lower their toxicity and hydrophobicity and for purification purposes was only partly successful and will need further improvement.
- ITC is an excellent proposition, however, we would need pure samples equilibrated in the exact same buffer at high concentration of gIIA1/gIIA2 peptides.

7) Do you have an idea of how much gallocin A is produced by the WT strain?

Unfortunately, without purified GIIA1 and GIIA2, we cannot determine gallocin A concentration in the supernatant of the WT strain. However, what we do know is that production of gallocin A is a highly regulated process and sensitive to quorum-sensing as demonstrated in our previous study (Proutière et al., mBio, 2021).

8) You mention that the WalRK mutants exhibit abnormal morphology, this is clearly shown in Fig. 7 and is well documented in the literature for other Streptococci. Could you provide enumeration of the chain length in SGM WT vs. these resistant mutants, i.e. are they statistically different? One of two of your more severe mutants would be acceptable.

We would like to share with the reviewer the chaining phenotype observed below by flow-cytometry on 10,000 bacteria for RSM1, RSM12 and RSM14. However, we feel that this quantification is a bit out of scope and that does not add much to the qualitative data presented in Fig. 7.

9) You mention in your methods that your control SGM WT strain had mutations compared to the reference sequence, and the RSM mutants did as well. Are these background mutations shared between WT and RSM mutants? Do you believe this is due to strain mutation/adaptation over time in the laboratory, as has been observed for other species?

Indeed, some of these mutations which were present in both WT and resistant mutants could indicate strain evolution over time in the laboratory. However, we do not think that these mutations are the results of strain adaptation but rather due to different sequencing methods: PacBio- long-reads for the reference SGM genome vs Illumina- short reads for SNP identification in the RSM mutants.

Other minor comments:

-Some of the pictures in Fig. S2 are quite pixelated. Could you provide higher resolution pictures for these? Also it would benefit to have multiple images, although I know this could make the figure quite busy and I don't think is absolutely necessary.

A low-resolution merged PDF containing both manuscript and figures was provided at this first stage of submission. Naturally, we will provide higher resolution image in TIFF or PNG format for all the figures. This figure is already quite busy so we will not provide multiple images.

-In Figure S5, control is spelled in French. Please change to English.

-In Line 254, I believe the authors meant to refer to Table 2 instead of Table 1.

Thank you for noticing, we changed it.

-Figure 4C, please outline what statistical test is used to compare the conditions.

Asterisks represent statistical differences relative to WT strain UCN34 with ***p < 0.001 as assessed by using two-way ANOVA in GraphPad Prism version 9.

-Figure S5 is a little difficult to interpret, Could you present in a slightly different way to make the areas of mutation larger?

-Please add scale bars to Figure 7 for the microscopy.

Ok

-Line 423-OD is spelled incorrectly.

Thank you for noticing, we changed it.

-Can you outline how many times each of the experiments were performed in the figure legends?

Ok

Reviewer #2 (Public repository details (Required)):

Bacterial genome data

Reviewer #2 (Comments for the Author):

General comments:

The authors studied the function of gallocin A, a bacteriocin of *Streptococcus gallolyticus* subsp. *gallolyticus*. The aim of the study is to characterize the mode of action and the mechanisms of resistance for gallocin A. The authors can demonstrate lipid bilayer disruption by gallocin A and they show by genetic approaches that gallocin A is a two peptide

bacteriocin and that GIP functions as an immunity protein. The main novelty of the manuscript are the disulfide bonds proposed to be important for the bacteriocin activity. This hypothesis is confirmed by a loss of bacteriocin activity upon treatment with reducing agents and the detection of a thioredoxin domain in a gene of the bacteriocin cluster that upon knockout causes diminished bacteriocin activity. To substantiate this observation a mutation of cysteine residues of gallocin A should be performed. Some information about the concentration at which gallocin A is killing bacterial target strains should also be included in the manuscript. Analysis of gallocin A resistant mutants suggest an involvement of the cell wall structure in the resistance mechanism but could not identify a specific protein receptor.

We would like to thank Reviewer 2 for carefully reading our manuscript and for her/his comments.

As detailed in the answer #4 Reviewer 1 and pasted below

“We agree with the reviewer that it would be a very good experiment to prove the importance of the cysteine residues in the gallocin A peptides. However, we are missing a tool to detect gallocin A peptides. All our attempts to tag the gallocin peptides abrogated the peptide activity. Thus, we have no simple way to check if the mutated GIA1/GIA2 peptide would be produced to the same quantity and not degraded in the presence of this mutation. Moreover, constructing mutants in *Streptococcus gallolyticus* is still quite challenging and time-consuming. We thought about introducing the mutations on the complementation plasmid, however since complementation is already only partial with the wild-type gene, it will be difficult to get unambiguous results”.

Unfortunately, without purified GIA1 and GIA2, we cannot determine gallocin A concentration in the supernatant of the WT strain. Thus, it is difficult to estimate the concentration of gallocin A needed to kill target bacteria.

Specific comments:

1 Page 2 line 55

SGG is not only associated with asymptomatic colon tumors, please modify.

We have removed the word “asymptomatic” from the sentence line 2. We hope that this answers your comment.

2 Page 12 line 297-304

Here the concentrations at which gallocin is active should be mentioned and that the spectrum of activity against closely related species is in line with previous investigations should be added.

As stated above, in the absence of purified peptides constituting gallocin A, we cannot determine the concentration of gallocin A. We have used in all our assays filtered bacterial supernatant of WT and mutant strains.

3 Page 24 line 624

Legend of Fig 4A Please provide information on the concentration of nisin that was used in this assay.

Thank you for pinpointing this omission. We have used nisin at 25 µg/mL and have added this information in Fig4 ' legend.

4 Fig. 2:

While the sensitivity of *S. agalactiae* has been tested. What about other pathogenic streptococci, like *S. pyogenes* or *S. dysgalactiae* subsp *equisimilis*?

We have tested two *S. pyogenes* isolates and two *S. dysgalactiae* subsp *equisimilis*, and all of them were found to be susceptible to gallocin A. We have added one representative plate of each specie in Fig. S2.

End of Authors reply

Finally, we would like to thank the two Reviewers for their comments and suggestions that we believe helped us to improve the quality of the manuscript.

We hope that the revised manuscript will be found acceptable for publication.

Sincerely,
The Authors

February 16, 2023

Dr. Shaynoor Dramsi
Institut Pasteur
28 rue du Dr Roux
Paris
France

Re: Spectrum05085-22R1 (Gallocin A, an atypical two-peptide bacteriocin with intramolecular disulfide bonds required for activity)

Dear Dr. Shaynoor Dramsi:

Your manuscript has been accepted, and I am forwarding it to the ASM Journals Department for publication. You will be notified when your proofs are ready to be viewed.

Sincerely,

Christopher LaRock
Editor, Microbiology Spectrum
